# Perilla Fruit Oil-Fortified Soybean Milk Intake Alters Levels of Serum Triglycerides and Antioxidant Status, and Influences Phagocytotic Activity among Healthy Subjects: A Randomized Placebo-Controlled Trial

**DOI:** 10.3390/nu14091721

**Published:** 2022-04-21

**Authors:** Pimpisid Koonyosying, Winthana Kusirisin, Prit Kusirisin, Boonsong Kasempitakpong, Nipon Sermpanich, Bow Tinpovong, Nuttinee Salee, Kovit Pattanapanyasat, Somdet Srichairatanakool, Narisara Paradee

**Affiliations:** 1Oxidative Stress Cluster, Department of Biochemistry, Faculty of Medicine, Chiang Mai University, Chiang Mai 50200, Thailand; pimpisid.k@cmu.ac.th; 2Department of Family Medicine, Faculty of Medicine, Chiang Mai University, Chiang Mai 50200, Thailand; wkusiris@gmail.com (W.K.); dr.boonsong@gmail.com (B.K.); nipon.s@cmu.ac.th (N.S.); 3Division of Nephrology, Department of Internal Medicine, Faculty of Medicine, Chiang Mai University, Chiang Mai 50200, Thailand; jingprit@hotmail.com; 4Program of Food Production and Innovation, Faculty of Integrated Science and Technology, Rajamangala University of Technology Lanna, Chiang Mai 50300, Thailand; bowtinpovong@rmutl.ac.th (B.T.); nattinee@rmutl.ac.th (N.S.); 5Office of Research and Development, Faculty of Medicine and Siriraj Hospital, Mahidol University, Bangkok 10700, Thailand; kovit.pat@mahidol.ac.th

**Keywords:** *Perilla frutescens*, fruit oil, ω3-PUFA, acceptance, antioxidant, health

## Abstract

This study aimed to develop perilla fruit oil (PFO)-fortified soybean milk (PFO-SM), identify its sensory acceptability, and evaluate its health outcomes. Our PFO-SM product was pasteurized, analyzed for its nutritional value, and had its acceptability assessed by an experienced and trained descriptive panel (*n* = 100) based on a relevant set of sensory attributes. A randomized clinical trial was conducted involving healthy subjects who were assigned to consume deionized water (DI), SM, PFO-SM, or black sesame-soybean milk (BS-SM) (*n* = 48 each, 180 mL/serving) daily for 30 d. Accordingly, health indices and analyzed blood biomarkers were recorded. Consequently, 1% PFO-SM (1.26 mg ALA rich) was generally associated with very high scores for overall acceptance, color, flavor, odor, taste, texture, and sweetness. We observed that PFO-SM lowered levels of serum triglycerides and erythrocyte reactive oxygen species, but increased phagocytosis and serum antioxidant activity (*p* < 0.05) when compared to SM and BS-SM. These findings indicate that PFO supplementation in soybean milk could enhance radical-scavenging and phagocytotic abilities in the blood of healthy persons. In this regard, it was determined to be more efficient than black sesame supplementation. We are now better positioned to recommend the consumption of PFO-SM drink for the reduction of many chronic diseases. Randomized clinical trial registration (Reference number 41389) by IRSCTN Registry.

## 1. Introduction

The *Perilla frutescens* var. nga-keemon plant (known by its Japanese name “shiso”) is cultivated in the upper northern provinces of Thailand, specifically Chiang Mai, Nan, and Maehongsorn Provinces. It is an edible fruit that is commonly used as an additive in rice, cookies, and biscuits. *P. frutescens* oil (PFO) is abundant with omega-3 polyunsaturated fatty acids (ω3-PUFA), particularly α-linolenic acid (ALA) at 54–65% (*w*/*w*) of the total fatty acid content [1,2,3]. In the human body, ALA is metabolized into eicosapentaenoic acid (EPA), docosahexaenoic acid (DHA), and eicosanoids, all of which are known to exert anti-inflammatory, anti-thrombotic, and various neuroprotective effects [4,5,6,7]. With regard to its anti-atherogenic properties, perilla oil has been observed to lower plasma lipids and decrease the size of fatty-streak lesions in apolipoprotein-E knockout mice [8]. Accordingly, these properties can improve various cardiovascular and immune functions [9,10,11]. Importantly, PFO is also comprised of lipophilic antioxidants, such as vitamin E, tocopherols, and tocotrienols [3,12,13]. Recently, we have determined that PFO possesses analgesic, anti-inflammatory, and anti-ulcerative properties [14], all of which can improve red blood cell (RBC) indices and protect against carbon tetrachloride-induced hepatotoxicity in animal models [3]. Moreover, the ethanolic extract of *P. frutescens* fruits is abundant with phenolics and flavonoids that have been observed to exhibit anti-inflammatory activities in tumor necrosis factor-alpha (TNF-α)-induced endothelial (EA. hy926) cells. Furthermore, this extract can also reduce oxidative stress and lipid peroxidation in human hepatocellular carcinoma (HuH7) cells [15,16].

At present, the food industry is focusing its efforts on developing new, healthy, and functional plant-based beverages. In fact, soybean milk (SM) possesses the highest protein content (35–50%, *w*/*w*) when compared with other kinds of milk [17]. Nutritionally, SM contains fatty acids, vitamins, minerals, antioxidants, and isoflavones [18,19]. Accordingly, it is a suitable option for people who are allergic to cow’s milk (lactose intolerance). Importantly, the isoflavones present in plant-based beverages have been found to effectively prevent various postmenopausal syndromes, prostate cancer, and osteoporosis [20,21]. Additionally, the regular intake of SM protein can lower serum triglyceride (TG) and low-density lipoprotein-cholesterol (LDL-C) levels [22]. Hence, there is an important need for the development of new functional dairy alternatives. This would include ω3-PUFA rich soybean milk. Sesame oil is an edible oil pressed from *Semen sesame* and *Sesamum indicum* seeds. It is known to be substantially abundant with tocopherols (particularly γ-tocopherol) and phenolic compounds. The oil exhibits interesting nutritional and biological properties [23]. In our research, we have fortified it with SM and preserved its shelf-life using high temperature treatment (UHT) in order to meet the needs of consumers [24]. Pasteurization is a method of food processing aimed to eliminate pathogenic bacteria, preserve most temperature-sensitive compounds (such as proteins and bioactive ingredients), and regulate the activity of mother’s milk [25,26,27]. Accordingly, sensory properties and consumer acceptance scores of product prototypes are required to optimize sensory quality based on the nine-point verbal hedonic box scale [28,29,30]. The fortification of PFO in SM has made it a multi-purpose drink that combines bioactive and nutraceutical properties. Therefore, it has received a significant amount of attention from consumers. We conducted our research in order to develop a pasteurized PFO-SM drink, identify its sensory characteristics and level of consumer acceptance, and to clinically investigate the relevant health outcomes among healthy participants after consumption of our PFO-SM product.

## 2. Materials and Methods

### 2.1. Chemicals and Reagents

Butylated hydroxytoluene (BHT), 2,7-dichlorodihydrofluorescein diacetate (DCFH-DA), tetramethoxypropane (TMP), thiobarbituric acid (TBA), trichloroacetic acid (TCA), 2-(4-iodophenyl)-3-(4-nitrophenyl)-5-(2,4-disulfophenyl)-tetrazolium monosodium salt (WST-1), and 2,2-diphenyl-1-picrylhydrazyl (DPPH) were obtained from Sigma-Aldrich Chemical Company (St Louis, MO, USA). R-Phycoerythrin (PE)-conjugated anti-CD13 antibody was purchased from BD Biosciences, San Jose, CA, USA. Hydrogen peroxide (30%) was purchased from Merck KGAA, Darmstadt, Germany. *Bis*-carboxyethyl-carboxyfluorescein pentaacetoxymethylester (BCECF-AM) was purchased from Thermo Fisher Scientific Inc, Branchburg, NJ, USA. Food ingredients (food grade), including red sugar (Mitr Phol Sugar Group, Bangkok, Thailand), table salt, and emulsifier/stabilizer (Palsgaard^®^ RecMilk 121, Palsgaard A/S Company, Juelsminde, Denmark), were purchased from a local grocery store located in Mueang Chiang Mai, Chiang Mai, Thailand. Glutathione assay kit was purchased from Sigma-Aldrich Chemicals Company, St Louis, MO, USA. SOD determination kit was purchased from Sigma-Aldrich Chemie, Buchs, Switzerland.

### 2.2. Soybean Milk Drinks

#### 2.2.1. Preparation of Pasteurized SM and PFO-SM Drinks

The fruits of Thai *Perilla frutescens* plants (Thai local name “Nga-keemon”) were harvested from agricultural fields located in Wienghang District, Chiang Mai, Thailand and kept in desiccated plastic bags at ambient temperatures for a number of days. They were mechanically pressed to release PFO using a hydraulic press (Zhengxi Company, Chengdu, China) and chemically analyzed in our laboratory at the Department of Biochemistry, Faculty of Medicine, Chiang Mai University [3,16]. Soybeans (*Glycine max* (Linn.) Merrill) were bought from a local grocery store located in Mueang Chiang Mai, Chiang Mai Province, Thailand. Washed soybeans were soaked in deionized water (DI, 1 kg/5 L) for 24 h, and blended homogenously at 60–70 °C using a soybean milk maker, filtered, boiled at 80 °C for 15 min, and cooled down. Finally, the SM was pasteurized using a pasteurizing machine at 80 °C for 15 min and packed in plastic bottles (180 mL/bottle) with the generous assistance of Hauykaew Diary Factory, Mueang Chiang Mai. The product was then stored in a refrigerator at 4 °C for up to 14 days until it was used in our study. PFO-SM was prepared according to our established procedure [14] by mixing SM (93%, *v*/*v*) with PFO (1%, *v*/*v*) containing 0.70 mg ALA) and 0.26 mg linoleic acid (LA), red sugar (5%, *w*/*v*), table salt (0.1%, *w*/*v*), and emulsifier/stabilizer (0.125%, *w*/*v,* Palsgaard^®^ RecMilk 121, Palsgaard A/S Company, Juelsminde, Denmark). The PFO-SM was then homogenized using a two-stage homogenizer. It was then pasteurized and packed in plastic bottles according to the process described above. Finally, the PFO-SM was pasteurized using a pasteurizing machine set at 80 °C for 15 min. The product was then packed in plastic bottles (180 mL/bottle) and stored in a large refrigerator at 4 °C for up to 14 days until it was used in our study.

#### 2.2.2. Analysis of Nutritional Facts

Pasteurized SM and 1% PFO-SM samples were randomly selected for determination of nutritional fact values at the Central Laboratory (Chiang Mai), Maerim, Chiang Mai, Thailand using the previously established methods. Total calories, fat calories, and total carbohydrate contents in the drinks were determined using the established methods of the International Officers and Committees 1993 [31,32]. Amounts of dietary fiber, protein, total sugars, total saturated and unsaturated fats; cholesterol, and vitamins were measured according to the methods prescribed by the Official Methods of Analysis (AOAC), 21st Edition (2019), AOAC International, Rockville, MD, USA [32,33,34]. Furthermore, elements, including iron (Fe), sodium (Na), and calcium (Ca), were stoichiometrically analyzed using the inductively coupled plasma/mass spectrometry (ICP/MS) technique according to the AOAC, for which the lowest limit of detection (LOD) was observed to be 1 mg/kg [35,36].

#### 2.2.3. Sensory Acceptability Test

Scores of the PFO-SM drink were evaluated by semi-trained sensory panelists who were familiar with the products. A nine-point hedonic scale test, as described by Meilgaard et al. [37], was used to evaluate the drinks. The panelists were comprised of healthy subjects (100 in total) who lived in Chiang Mai Province, Thailand. Members of the panel were divided into two groups: Group 1 received the 0.5% (*v*/*v*) PFO-SM drink and Group 2 received the 1.0% (*v*/*v*) PFO-SM drink. Sessions were conducted in an air-conditioned sensory test laboratory under conditions of controlled lighting and ambient temperatures (22–25 °C) at the Program of Food Production and Innovation, Faculty of Integrated Science and Technology, Rajamangala University of Technology Lanna, Doi Saked Campus, Chiang Mai, Thailand. After a period of training on the use of these products, the PFO-SM drinks were blind-tested in triplicate by the panel. In the tests, the order in which the drinks were presented to the participants was random. At each session, the drinks were served in transparent plastic cups coded with 3-digit numbers. The subjects were then asked to score the degree of intensity in terms of the sensory attributes using a 9-point hedonic scale ranging from 1 = ‘dislikes very much’ to 9 = ‘likes very much’. In order to establish an accurate assessment of the level of acceptance of the consumers, we used a 9-point hedonic scale to assess the degree of approval among voluntary consumers (*n* = 100). Each consumer was invited to test each PFO-SM sample (100 mL presented in random order and coded with three-figure random numbers). They were then asked to score their degree of acceptability with respect to appearance, taste, and their overall liking using a nine-point verbal hedonic box scale with scores ranging from 1 = ‘dislikes very much’ to 9 = ‘likes very much’.

#### 2.2.4. Black Sesame-Soybean Milk

Black sesame-fortified soybean milk (BS-SM) was purchased from a Tesco Supermarket in Muang, Chiang Mai Province, Thailand. The nutritional facts of the BS-SM drink (230 mL/carton) were established according to the manufacturer’s information (Table 1).

### 2.3. Clinical Study

#### 2.3.1. Study Design and Setting

The study protocol was approved by the Ethical Committee for Human Study at the Faculty of Medicine, Chiang Mai University, Chiang Mai, Thailand (Research ID: 2829/Study code No.FAM-2558-02829). This single-center, randomized controlled demonstration trial was designed according to CONSORT guidelines and registered by the ISRCTN Registry Office, London, UK (number: ISRCTN11403535). The study was conducted at the Out-Patient Clinic, Department of Family Medicine, Faculty of Medicine, Chiang Mai University. All participating subjects were interviewed in order to meet the designated inclusion criteria. For the four independent groups, sample size was calculated using the following formula:*n* = 2 × [(Z_α_ + Z_β_)PQ]^2^/(P_t_ − P_c_)^2^(1)
where *n* = sample size, Z = a constant, P = 0.23, Q = 0.77, P_t_ = 0.66, and P_c_ = 0.20.

Accordingly, the sample size was comprised of forty-four subjects in each of the groups, along with an additional 10% to compensate for any subjects who may have withdrawn or who would be unable to continue taking part in the study.

#### 2.3.2. Subject Preparations

Volunteers were made up of men and women aged 20–60 years old who were not infected with any serious diseases. For the purposes of inclusion, all volunteers who agreed to participate throughout the course of the study, were able to communicate in Thai and provided their informed written consent. Subjects were also deemed not to be allergic to soybeans or perilla nutlets and were not currently participating in any vitamin, supplementary food, or any other controlled diet programs. For the purposes of exclusion, subjects who received drugs that may have interfered with the results were declined consent. Subjects were further excluded if they were allergic to soybeans and/or perilla fruits, or exhibited adverse effects during the course of the study.

#### 2.3.3. Study Interventions

One hundred and ninety-two subjects were recruited and randomly divided into four groups (*n* = 48 each) who were given the DI, SM, BS-SM, or PFO-SM drinks. All of the subjects willingly consumed the drinks (180 mL/bottle each) after breakfast and dinner twice daily for 30 days (Figure 1). During the course of the study, all subjects maintained the consumption of their usual diets and continued engaging in routine physical activities. In terms of prohibition, they were asked to refrain from smoking, drinking alcohol, and consuming vitamins or any other dietary supplements. Overnight-fasting blood samples were withdrawn from venous veins and collected in plain tubes and tubes coated with ethylenediaminetetraacetic acid (EDTA) at the onset of the experiment and after consumption of the drinks for 30 days. In terms of physical examinations, body weight (BW), body mass index (BMI), systolic blood pressure (SBP), and diastolic blood pressure (DBP) were recorded at the same periods of time for each participant.

#### 2.3.4. Analysis of Blood Biomarkers

*Hematological Parameters:* EDTA blood was analyzed for complete blood count (CBC) including RBC, white blood cell (WBC), and platelet (PLT) indices using an Automated cell counter/analyzer (Beckman Coulter Life Sciences, Indianapolis, NJ, USA) according to the manufacturer’s instructions.

*Biochemical Parameters:* Serum was separated from clotted blood samples by centrifugation at 3000× *g* for 10 min and kept frozen at −20 °C until analysis. Biochemical parameters, including blood urea nitrogen (BUN), creatinine (CRE), aspartate aminotransferase (AST), alanine aminotransferase (ALT), alkaline phosphatase (ALP), total cholesterol (TC), triglycerides (TG), high-density lipoprotein-cholesterol (HDL-C), and low-density lipoprotein-cholesterol (LDL-C), were analyzed using an Automated Randox Daytona Clinical Chemistry Analyzer (Randox Laboratories Limited, County Antrim, UK) according to the manufacturer’s instructions.

*Phagocytotic Activity:* Phagocytotic activity in the blood was measured according to the previously described method [38]. EDTA blood samples were diluted with phosphate buffered saline (PBS) and incubated with PE-conjugated anti-CD13 antibody for 30 min. After that, the cells were washed, suspended in PBS, and incubated with BCECF-AM-labeled *Candida albicans* (dilution 1:1) at 37 °C for 1 h. Then, 0.1% paraformaldehyde was added and the fluorescence intensity (FI) of PE and BCECF-AM was measured at excitation wavelengths of 488 nm, while emission values were detected at 530 nm for PE and 575 nm for BCECF-AM using a flow cytometer (BD FAC™, BD Biosciences, Franklin Lakes, NJ, USA) [39].

*Erythrocyte ROS:* ROS in RBC was detected using the DCFH-DA/flow cytometry method [40]. Ten EDTA blood samples were randomly selected from the total subjects in each group and centrifuged at 1500× *g* for 20 min to separate RBC from the plasma. Packed RBCs were washed twice with PBS and suspended in PBS to achieve 5% RBC suspension. The RBC suspension was incubated with 10 µM DCFH-DA fluorochrome for 30 min in the dark and challenged with 125 µM (*v*/*v*) hydrogen peroxide for 1 min in the dark, while FI was measured using a flow cytometer at an excitation wavelength of 485 nm and an emission wavelength of 530 nm.

*Thiobarbituric Acid-Reactive Substances (TBARS):* Serum concentrations of TBARS, such as malondialdehyde (MDA), were measured using the colorimetric method established by Masowicz and colleagues with slight modifications [41]. Briefly, the serum and standard TMP solution (80 µL) were mixed with 0.2% BHT (10 µL). Next, 0.44 M meta-phosphoric acid (240 µL) and 0.6% (*w*/*v*) TBA (160 µL) were added to the mixture, which was then incubated at 90 °C for 30 min and cooled down in an ice bath at 4 °C for 10 min. After that, *n*-butanol (300 µL) was added into the subsequent mixture and centrifuged at 3000 rpm for 15 min. Finally, the supernatant was measured in terms of absorbance (A) at a wavelength of 540 nm.

*Superoxide Dismutase (SOD):* SOD activity was determined using Dojindo’s highly water-soluble tetrazolium salt with WST-1 as an electron acceptor, according to the procedure described by the manufacturer [42]. Briefly, serum and deionized water (DI) (20 µL) were mixed with the WST working solution (200 µL). It was then incubated with xanthine/xanthine oxidase working solution at 37 °C for 20 min. The A value was then photometrically measured at 440 nm. Since the A value of WST-1 formazan was directly proportional to the amount of superoxide anion, the SOD activity (% inhibitory rate) was calculated using the following equation:% Inhibition of SOD activity = 100 × (A_serum_ − A_DI_)/A_DI_(2)

*Reduced Glutathione (GSH):* Serum GSH concentration was determined based on the non-enzymatic reduction of 5,5′-dithiobis (2-nitrobenzoic acid) substrate to yellow-colored 5-thio-2-nitrobenzoic acid product giving a maximal absorption value of 450 nm [43]. The assay procedure was conducted following the manufacturer’s instructions.

*Total Antioxidant Capacity (TAC):* The TAC value in the serum was measured using DPPH radical scavenging assay [44]. Serum samples were deproteinized with an equal volume of 12.5% (*w*/*v*) TCA for 10 min and centrifuged at 12,000× *g* at a temperature of 4 °C for 10 min. The supernatant (20 µL) or DI was mixed with 10 mM DPPH containing radical (DPPH^•^) solution (25 µL) and methanol (970 µL). It was then incubated for 3 min, and the A value was photometrically measured at 517 nm. Serum TAC was expressed as percentage of scavenging activity of produced DPPH^•^, which was then calculated using the following equation:Scavenging activity (%) = (1 − A_serum_/A_DI_) × 100(3)

### 2.4. Statistical Analysis

All statistical analyses were performed using the SPSS Program version 18.0 (IBM, Armonk, NY, USA licensed by Chiang Mai University) and expressed as values of mean ± standard deviation (SD). Descriptive analysis for the results of the sensory acceptance assessment was employed to describe voluntary consumer characteristics as well as all variables related to the PFO-SM drink, product consumption, and sensory analysis. Evaluation of product acceptance by the panelists was converted into scores of −3, −1, 1, and 3, so that there were equal intervals between adjacent scores throughout the scale. When all variables were not distributed normally (Kolmogorov–Smirnov test), nonparametric analysis was performed. Differences observed at different times in the same group were analyzed with the parametric paired samples obtained from the paired Student’s *t*-test or the non-parametric Wilcoxon test. Any differences observed between the two groups were analyzed using the parametric independent samples of the *t*-test or the non-parametric Mann-Whitney U test. Differences between the four groups were analyzed using parametric analysis of variance (ANOVA), followed by post hoc Tukey’s HSD, Dunnett T3, or the non-parametric Kruskal-Wallis test. Accordingly, *p* values of <0.05 were considered statistically significant, as were ^a^
*p* values of <0.05, when comparisons were made with baseline (D0) data in the same group, using a parametric paired Student’s *t*-test; ^b^
*p* values of <0.05 when comparisons were made with baseline (D0) data in the same group using non-parametric Wilcoxon test; ^c^
*p* values of <0.05 when comparisons were made between changes of the four groups in the brackets using parametric one-way ANOVA followed by post hoc Tukey’s HSD or Dunnett T3 tests; and ^d^
*p* values of <0.05 when comparisons were made between changes of the four groups in the brackets using non-parametric Kruskal-Wallis test; ^e^
*p* values of <0.05 when comparisons were made between male and female subjects in the same group using parametric independent samples of the Student’s *t*-test; and ^f^
*p* values of <0.05 when comparisons were made between male and female subjects in the same group using non-parametric Mann-Whitney U test.

## 3. Results

This section has been divided by subheadings in order to provide a concise and precise description of the experimental results, their interpretations, as well as the experimental conclusions that can be drawn.

### 3.1. Pasteurized PFO-SM Drink

Results of the nutritional analysis of the SM, 1% PFO-SM, and BS-SM drinks per 180 mL servings are provided in Table 1. Interestingly, our SM and PFO-SM drinks were associated with fewer total calories (90%) and fat calories (44.4% and 66.7%, respectively) than the BS-SM drink. Additionally, the two products contained less total fat (60% and 40%, respectively), trans fat, and cholesterol when compared to the BS-SM drink. Importantly, the PFO-SM product had a larger amount of PUFA than the SM product, which had been fortified with 1.26 mg of ALA and 0.47 mg of LA. Likewise, the SM and PFO-SM drinks contained higher carbohydrate and sugar contents than BS-SM, but fewer amounts of dietary fiber and protein than the BS-PFO drink. Importantly, our two products contained much less sodium content than the BS-SM drink. 

### 3.2. Sensory Acceptability Assessment

Herein, we introduced our PFO-fortified soybean milk, offering significant health benefits exerted by 3ω-ALA abundance. In addition, the drink was evaluated in terms of consumer satisfaction and acceptability. As shown in Table 2, a total of one hundred healthy consumers participated in the assessment of the 1% PFO-SM product. The group was comprised of fifty-two female subjects and forty-eight male subjects who were aged 18–55 and who were involved in a variety of professions and roles, including university students (70%), lecturers (15%), back officers (10%), and others (5%). We found that most of the consumers participating in the test preferred a product price within a range of 15–20 baht (88%) to that of 21–25 baht (12%). After consumption, 68% of the evaluators stated that they intended to buy the product, 15% of them did not want to buy the product, and the remaining 17% were unsure. Logically, the product preference was determined by price (65%), taste (24%), brand (3%), and its health benefits (8%).

According to the sensory acceptance evaluation (Table 3), the subjects expressed nearly equal preferences for the 0.5% and 1.0% PFO-SM products in terms of color, odor flavor, sweet taste, additive taste, and texture on the 9-point hedonic scale assessment. Though the 0.5% PFO-SM drink was found to be accepted slightly more often than the 1% PFO-SM drink (*p* > 0.05), the 1% PFO-SM drink was chosen to be used in the clinical trial because it contained greater ALA content, improved flavor and taste, and would likely exert greater beneficial effects on human health. The resulting figures were then compared with those of a commercial black sesame-fortified soybean milk product.

The data shown in Table 3 demonstrate that consumer satisfaction scores, with regards to color, aroma, taste, texture, and overall satisfaction were not significantly different when comparisons were made between 0.5% PFO-SM and 1% PFO-SM drinks. Sixty-eight percent of the consumers were slightly to moderately satisfied with the two products. Additionally, consumers aged 15–35 years old were predominantly satisfied with 0.5 and 1% PFO-fortified soybean milk products. Moreover, these same consumers stated that they were willing to pay 15–20 baht for these tested products (180–230 mL serving). 

### 3.3. Subject Information

A total of one-hundred and ninety-two participants were enrolled in this study, including thirty-seven men (19.3%) and one-hundred and fifteen women (80.7%) aged 33.3 ± 11.4 years, who were classified into groups according to age, as follows: 20–29 years (*n* = 84, 43.7%), 30–49 years (*n* = 85, 44.3%), and 50-60 years (*n* = 23, 12.0%). Personal details and specific information, including gender, age, and health indices, of all the participants, and of those who had consumed DI, SM, PFO-SM, or BS-SM drinks for 30 days, are shown in Table 4. With regard to gender, in accordance with our inclusion criteria, there were more female subjects than male subjects in all four groups; however, no significant differences were observed among these three groups. With an effective method of randomization, no significant differences were observed in terms of age, BW, BMI, SBP, and DBP of the participants of all four groups at the beginning of the study. Following consumption of the test products for 30 days, all the values were still within normal ranges and were not significantly changed. Hence, consumption of PFO-SM did not influence the health index values when compared to consumption of SM or BS-SM alone.

### 3.4. Effect on Hematological Parameter Levels

The effects of DI, SM, PFO-SM, and BS-SM intake for 30 days on levels of RBC, WBC, and PLT indices are presented for all subjects as well as for male and female subjects. As is shown in Table 5a, most of the RBC, WBC, and PLT parameters for all four groups on days 0 and 30 were within normal ranges, of which those of the PFO-SM groups on day 30 were not significantly altered when compared with the levels on day 0. Following consumption of the drinks for 30 days, levels of RBC numbers, and Hb and Hct values significantly decreased in the SM group, but not in the PFO-SM and BS-SM groups. However, levels of the other RBC indices were not significantly altered in all four groups according to the study time, while these levels were not found to be different between the groups. Similarly, levels of WBC and PLT parameters in all four groups were neither significantly altered by consumption of the drinks nor different between the groups.

With regard to gender, these parameter values were reported separately for male and female subjects (Table 5b). Significant differences of RBC values were observed, as well as for Hb, Hct, MCV, MCH, RDW, and WBC values, while % differential monocytes and eosinophils, PDW, and Pct values were observed for both male and female subjects before and after consumption of the drinks. Apparently, there were no significant changes in all of the hematological parameters observed in the subjects for both male female subjects who consumed the drinks. Accordingly, female subjects did not influence the outcomes even, though larger numbers of females were enrolled in this study. Exceptionally, changes in the differential monocytes were −2.3 ± 2.3% in male subjects and 0.3 ± 2.5% in females (^g^
*p* < 0.05), while no significant differences were observed between both male and female subjects.

### 3.5. Effect on Serum Levels on Biochemical Markers

As is shown in Table 6a, serum BUN, CRE, AST, ALT, and ALP values stayed within normal ranges in all four groups before and after consumption of the drinks for 30 days. Though consumption did not influence kidney function, the BS-PFO product was observed to decrease CRE levels significantly when compared to the baseline levels on day 0. Surprisingly, the PFO-SM product decreased the AST levels, while the BS-PFO product decreased both the AST and the ALP levels (*p* < 0.05). Notably, consumption of all four drinks for 30 days significantly elevated serum levels of TC, LDL-C, and TG but decreased serum levels of HDL-C. However, the PFO-SM product appeared to lower serum TG levels, while the SM product increased serum TG levels (^c^ *p* < 0.05). Moreover, there were significant changes in the TG, and TC levels when comparisons were made between the DI group and the SM group; along with changes in the CRE, AST, ALP, TG, TC, and LDL-C levels when comparisons were made between the SM group and the BS-SM group; and changes in the CRE, AST, ALT, ALP, TG, TC, and LDL-C levels when comparisons were made between the PFO-SM group and the BS-SM group. Similarly, there were neither significant differences in all biochemical parameters at any time during the course of the study (D0 and D30) nor significant changes after consumption of the drinks when comparisons were made between male and female subjects (Table 6b). 

### 3.6. Effect on Phagocytotic Activity

The results shown in Figure 2 indicate that phagocytotic activities increased slightly in the SM, PFO-SM, and BS-SM groups, whereas they were unchanged in the DI group after consumption of all drinks for 30 days. Interestingly, the change in phagocytotic activity of the PFO-SM group was significantly greater than that of the SM and BS-SM groups. It was observed that all the phagocytotic activity values recorded from male and female subjects were the same and were not significant even though there were more female subjects than males. 

### 3.7. Effects on Oxidative Stress and Antioxidant Activity

Levels of erythrocyte ROS and antioxidant biomarkers in the serum collected from subjects consuming the DI, SM, PFO-SM, and BS-SM drinks for 30 days are presented herein. According to our findings, erythrocyte ROS levels on day 30 were significantly lower than those on day 0 in the SM, PFO-SM, and BS-SM groups, while they were unchanged in the DI group. Surprisingly, the PFO-SM product was determined to be more effective than the SM and BS-SM products (^d^
*p* < 0.05) (Figure 3a). Likewise, serum MDA levels on day 30 were higher than those on day 0 in all four groups, for which differences were observed in the SM, PFO-SM, and BS-SM groups (Figure 3b). Moreover, serum GSH levels were found to have significantly increased in all four groups after consumption of the drinks, while alterations were not found to be significantly different among these groups (Figure 3c). However, serum SOD activity levels were decreased in all four groups, of which the decreases were observed to be significant in the SM, PFO-SM, and BS-SM groups (^b^
*p* < 0.05) after consumption of the drinks (Figure 3d). Furthermore, serum TAC levels were greatly increased after consumption of the PFO-SM (^a^
*p* < 0.05) and SM drinks for 30 days, whereas TAC levels decreased after consumption of DI (^a^
*p* < 0.05) and BS-SM (^b^
*p* < 0.05). Importantly, the alterations of TAC in the PFO-SM group were determined to be the most significant when compared with the BS-SM (^c^
*p* < 0.05) and SM (*p* > 0.05) groups (Figure 3e).

## 4. Discussion

Sesame seed oil is used primarily in salad oil, and has served as both a frying medium and a nutrient additive in beverages. In evaluating the quality of certain edible vegetable oils, the nutritional indexes of sesamin oil were reported with regard to total saturated fatty acids (SFAs), atherogenic fatty acids, including SFAs with 12, 14, and 16 carbon chains, monounsaturated fatty acids, and PUFAs such as LA and ALA [45]. Sesame oil is derived from the benne plant (*Sesamum indicum*) and is commonly consumed in African-American cuisine. This is the first study to focus on developing a functional pasteurized perilla fruit oil-fortified soybean milk and identify the relevant sensory characteristics, along with its level of consumer acceptance. Herein, we have developed a pasteurized PFO-fortified SM drink product that is rich in ALA and then identified its degree of sensory acceptance by consumers. According to the resulting evaluations, most of the consumers in the test expressed an interest in buying the product (68%). These consumers stated that they were willing to pay 0.45–0.60 US dollars (15–20 baht) for the 1.0% PFO-SM product. Assessments were made according to the price, taste, brand, and health benefits of the product. We then determined the outcomes of the consumer evaluations with regard to the product’s color, odor, flavor, sweet taste, additive taste, and texture. In comparison, the price of commercially available and high temperature manufactured black sesame-fortified soybean milk (180 mL/carton) is 0.28–0.30 US dollars (9.50–10 baht), which is 1.5 = 2 times cheaper than pasteurized PFO-SM. In the future, the production of PFO-SM beverages, on both commercial and industrial scales, will pull down the manufacturing costs and increase the product’s cost effectiveness as it relates to its potential benefits to human health. With regard to the nutritional facts, our pasteurized PFO-SM contains less calories, fats, and salts but does contain higher temperature-sensitive bioactive compounds than the BS-SM product. This is indicative of greater nutraceutical content and improved health benefits. Notably, the high temperature treatments (120–180 °C) associated with the manufacturing of sesame oil products via roasting, frying, and possibly sterilization can influence the PUFA ratios of C16:0, C18:0, C18:1, and C18:2 chains and significantly decrease the mineral content [46]. Dhibi and colleagues reported that fried sesame oil had higher amounts of atherogenic lipid products (such as trans fats and conjugated linoleic acid) than fresh sesame oil [47]. More importantly, a previous study has supported the determination that the pasteurization (62.5 °C for 30 min) of human milk did not affect total fat content and the percentage of compositions of saturated fatty acids, monounsaturated fatty acids, and ω3- and ω6-PUFA levels. However, sterilization (120 °C for 30 min) was found to significantly reduce the amounts of available fatty acids, such as LA and arachidonic acid (AA) [48,49]. Nevertheless, both pasteurization and sterilization methods induced oxidative degradation and increased amounts of propanal (or MDA) of 1% linseed oil-enriched dairy products through the chain-reaction peroxidation of the PUFA existing in linseed oil [50]. 

Though knowledge of consumer perception to food and drink products is growing rapidly at present, relevant knowledge and health outcomes established through the consumption of PFO-SM drinks are now being investigated using randomized clinical trials. Taken together, we chose to add 1% (*v*/*v*) PFO, rather than 0.5% (*v*/*v*) PFO, to SM, so that we would obtain a more concentrated PFO-SM drink (higher ALA abundance), greater flavor and taste (as described above), and improved biological and pharmacological effects [3]. In the present study, 1% PFO-SM drink was chosen to be tested among healthy subjects and the findings were compared with those of a basic SM product and a commercial BS-SM product. Likewise, the DI group was included in this study to confirm that the different outcomes possibly resulted from the consumption of either soybean milk or perilla oil, or a combination of the two. In our findings, no significant results were observed when comparisons were made between the DI group and the SM group. This suggested that the SM compositions did not influence, or change, the relevant health outcomes and levels of the blood biomarkers in the participants who consumed the drinks for 30 days. Indeed, we found that consumption of PFO-SM for 30 days lowered levels of serum triglycerides and red cell ROS, while increasing serum antioxidant activity in healthy subjects when compared to basic SM and a commercial BS-SM product. We have previously reported an abundance of unsaturated fatty acids, including ω3-ALA, ω6-linoleic acid, and ω9-oleic acid (57, 20 and 13% of total fatty acids, respectively); α- and β-tocopherols (2.5 and 4.95 mg%, respectively), γ-tocotrienol (4.37 mg%), and δ-tocopherol (1.28 mg%), in PFO. This would indicate its ability to exert protective effects on carbon tetrachloride-induced hepatotoxicity in rats [3]. In addition, PFO also exhibited analgesic and anti-inflammatory effects in rats [14]. In the human body, ALA is metabolized to PUFAs, such as EPAs and DHAs, while LA is a precursor for the biosynthesis of arachidonic acid (AA), via the process of catalysis by elongase and reductase [51,52]. Subsequently, both EPA and AA are further converted to the three main classes of eicosanoids, including prostaglandins, thromboxanes, and leukotrienes, which are known to exhibit biological and pharmacological functions in the human body [53]. Though ALA and LA are competitively metabolized using the same enzymes, eicosanoids obtained from ALA and LA act in opposing ways in the human body [54]. Accordingly, EPA-derived eicosanoids perform anti-inflammatory and anti-thrombotic activities, while AA-derived eicosanoids reveal pro-inflammatory and pro-thrombotic activities and vice versa [10,55]. Several studies have demonstrated that the intake of perilla oil increased the plasma levels of EPA and DHA and was associated with lower incidences of cardiovascular disease, while overconsumption of ω6-PUFA was highly associated with the prevalence of chronic inflammatory diseases and abnormal clotting [1,2]. Therefore, maintaining a balanced ratio of ω6:ω3 at 4–5:1 in one’s diet is important in preventing incidences of chronic inflammatory diseases [10,56]. Certainly, PFO would be recommended as a good resource for supplying ω3-PUFA within a well-balanced ratio of ω6:ω3 PUFA. 

Notably, there were more female subjects than male subjects (ratios of 2:1-4:1) in all four groups participating in this study. Nevertheless, no significant differences of the participants’ ages among the four groups and between the groups were observed. In the present study, the subjects were in good physical health and their weights were considered appropriate for their age and sex. Hematological and serum biochemical profiles were normal and there were no major differences or significant differences between the two groups in terms of sex distribution at the baseline and after consumption of the drinks. Consequently, any potential confounding factor pertaining to age or gender would not have influenced the outcomes of our study. Some previous studies have reported that sesame oil, rich in PUFAs, such as LA, inhibited the growth of HT-29-malignant human colon cancer cell lines and thymoma cells in C57BL/6 mice before and after being challenged with EL4 lymphoma cells [57,58]. Interestingly, after consumption of a sesamin oil-supplemented diet, LA is metabolized to ω6 dihomo-γ-linolenic acid product exhibiting antioxidant and anti-inflammatory activities and subsequent hepatoprotective effects against hepatic ischemia-reperfusion injury [59]. Surprisingly, sesame oil mixed with (10%, *v*/*v*) ALA-rich *Garden cress* oil revealed a significant decrease in serum levels of TC, TG, and LDL-C in rats [60]. Contrastingly, we found that the consumption of black sesame oil-fortified soybean oil (BS-SM) increased serum levels of atherogenic lipids, such as TC, TG, and LDL-C, in healthy human subjects. According to the lipid profiles, PFO-SM decreased the serum levels of TG significantly; while SM and BS-SM effectively increased serum TG levels. Though there were significant alterations in some biochemical parameter levels in the PFO-SM group, when compared with the BS-SM group, the changes were not deemed to be clinically relevant. In spite of small fluctuations, the values remained within normal clinical ranges. Soung and colleagues have revealed that post-menopause women consuming 60 mg isoflavone-supplement soy protein (5 g) drinks for 1 y showed an increase in RDW (a marker of reticulocytes) [61]. Interestingly, mild-asthmatic patients (11F, 4M; mean age, 61.0 y) who were fed a PFO salad (14.65 ± 1.41 g/d) for 4 w showed significant decreases in serum total cholesterol, LDL-C (*p* < 0.05), and triglyceride levels suggesting lipid-modulating effects [62]. Consistently, type 2 diabetic patients (20M, 30F, age 63.84 ± 9.67 y) who were orally administered PFO capsules for 6 months showed decreases in serum TC, LDL-C (*p* < 0.05), and TG levels [63]. It has been postulated that the ω3-ALA present in PFO could be a key ingredient in effectively decreasing the levels of blood lipids, particularly TG. This would possibly have occurred by the activation of fatty acid oxidation or/and competitive inhibition of intestinal lipid absorption. Hypothetically, the intake of ALA would activate plasma membrane fatty acid translocase (FAT/CD36) to translocate fatty acyl CoA across the membrane [64,65]. Moreover, consumption of essential ALA was found to increase maximal fat oxidation in athletes [66]. Several studies found that ω3-PUFA increased the expression of carnitine palmitoyl transferase I, which is known as the rate-limiting enzyme involved in fatty acid oxidation. However, ω3-PUFA also suppressed the expression of acetyl-CoA carboxylase known as the rate-limiting enzyme in fatty acid synthesis [67,68]. Furthermore, administration of cold-pressed perilla oil was observed to significantly reduce the expression of the peroxisome proliferator-activated receptor (PPAR)γ and fatty acid synthase (FAS) activities in incidences of high-fat diet-induced obesity in mice, leading to an inhibition of hepatic steatosis [69]. Consumption of PFO-SM by healthy volunteers for 30 days did not produce any changes in levels of hematological parameters or biochemical parameters, except for a significant decrease in serum AST. Consistently, our previous study has reported that oral administration of PFO for 90 d significantly decreased serum levels of ALT, ALP, and particularly AST in carbon tetrachloride-induced hepatotoxicity in rats, which would be indicative of its hepatoprotective effect [3].

Thies and colleagues demonstrated that healthy subjects in group A (4M, 4F), group B (5M, 3F), and group C (3M, 4F), who ingested the ALA blend (a mixture of palm, sunflower seeds, and flaxseed oil), DHA, and fish oil (a mixture of EPA and DHA) for 12 weeks were not associated with changes in total and differential WBC numbers or phagocytotic activity [70]. Our findings have revealed that PFO-SM consumption significantly enhanced phagocytosis when compared with SM and BS-SM, which could be attributed to bioactive phenolic compounds, but not 3ωPUFA *per se* [71].

With regard to oxidative stress and antioxidant capacity profiles, similarly, there were no significant differences between the two groups in terms of sex distribution at the baseline and after consumption of the drinks. Nevertheless, the two groups differed with regard to the efficacy of the PFO-SM drinks, although blood analysis excluded any possible confounding effect of sex differences in the outcomes of these parameters. Indeed, we are confident that PFO-SM exerted ROS-scavenging properties in RBC and plasma compartments, possibly by directly using conjugative double bonds to absorb the persisting ROS and/or to enhance antioxidant defense system in the body. Marinkovic and colleagues have demonstrated that the pasteurization process did not influence total antioxidative properties, but did change specific components and decreased the SOD and glutathione peroxidase activities of raw human milk per se [26]. In our study, we have observed a significant decrease in red cell ROS along with an increase in the serum antioxidant capacity among members of the PFO-SM group. We have demonstrated that PFO is abundant with lipid-soluble antioxidants, including tocopherols and tocotrienols [3]. Evidently, cold-pressed PFO was found to lower ROS levels in ultraviolet-induced normal human dermal fibroblasts [72]. Moreover, Lee and colleagues have illustrated that PFO exhibited a greater degree of DPPH radical-scavenging activity than oleic acid-rich olive oil and linoleic acid-rich corn oil [73]. Similarly, the administration of α-tocopherol for 3 months significantly reduced levels of red cell ROS and serum lipid-peroxidation products in β-thalassemia intermedia patients [74]. From a current randomized, double-blind clinical study, healthy elders who consumed PFO (0.88 g ALA rich)-fortified ponkan powder (2.91 mg nobiletin abundant) for 12 months revealed significantly higher cognitive index scores than the PFO group. This would suggest that a pro-cognitive effect is accompanied by increases in ALA and DHA levels in red cell membranes, serum brain-derived neurotropic factor levels, and biological antioxidant capacity [75]. Likewise, the supplementation of α- and γ-tocopherols for 6 weeks significantly decreased levels of plasma lipid peroxides and MDA in subjects diagnosed with metabolic syndrome [76]. Moreover, cold pressed-PFO treatment effectively suppressed oxidative stress and decreased SOD activity in PC12 cell cultures [77]. Surprisingly, the intake of dietary PFO increased serum antioxidant capacity and ALA levels in the RBC plasma membrane of healthy volunteers who were within an age range of 64-84 years old [78]. Taken together, the findings suggest that PFO-SM abundant with fat-soluble antioxidants, including tocopherols, tocotrienols, and PUFAs, could efficiently detoxify the ROS existing in blood components and prevent oxidative stress-related disorders.

Herein, we are attempting to introduce perilla fruit oil into the food and drink industry as a potential way to substitute PUFAs for the saturated fatty acids found in soybean milk. Importantly, it represents a cheaper plant-derived form of ω3-PUFA than deep-sea water fish oil. We would like to specifically highlight its potential health promoting benefits. In terms of limitations, PFO is hardly soluble in an aqueous SM base, so it would be very difficult to increase the amounts of ALA and other active ingredients as a way of achieving a greater degree of efficiency. Most of the subjects enrolled in this study were women who may have been affected by their menstrual cycles, resulting in blood loss and possible changes in red cell indices. Lastly, the study time was only one month, which may have been too short to observe considerable differences in these parameters. 

## 5. Conclusions

Pasteurized PFO-fortified soybean milk contains fewer calories and received highly satisfactory acceptance scores by the consumer panel. The findings of the randomized clinical study revealed that the drink did not influence health outcome values and most blood biomarkers. Indeed, the product lowered serum triglyceride levels, improved antioxidative properties in red cells and plasma compartments, and enhanced phagocytotic activity in healthy volunteers. This would suggest it has potential as an alternative healthy food for the Thai population. Overall, *Perilla frutescens* nutlet oil has been acknowledged as a suitable supplementary ingredient in soybean milk with a range of human health benefits. Further studies are warranted to clinically assess the effects of PFO-SM against certain oxidative stress-associated inflammatory, neurodegenerative, and chronic diseases.

## Figures and Tables

**Figure 1 nutrients-14-01721-f001:**
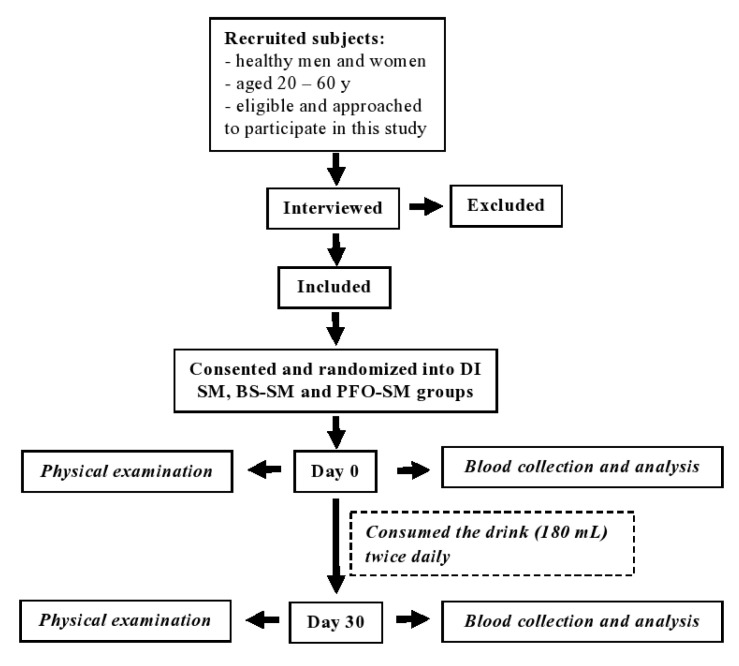
Overview of recruitment, randomization, and intervention protocols in the study.

**Figure 2 nutrients-14-01721-f002:**
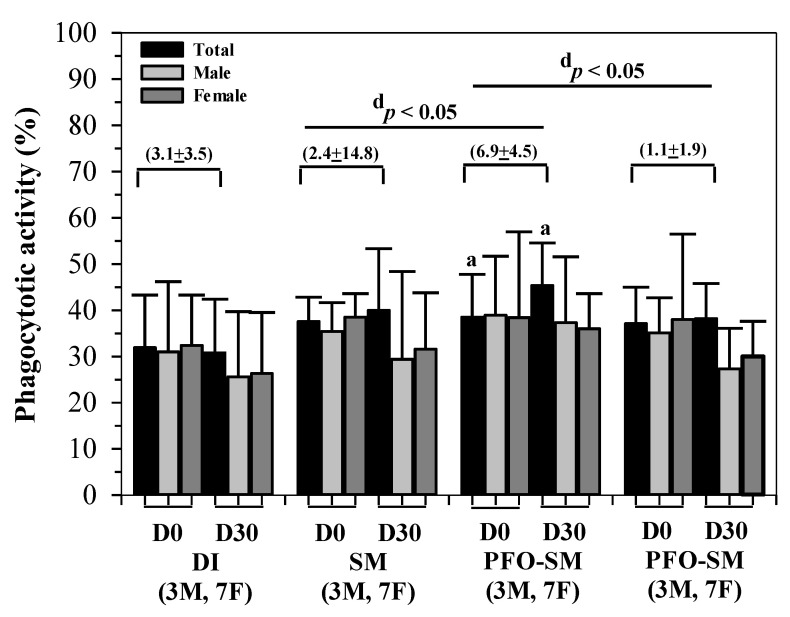
Phagocytotic activity of the PBMC product was determined from ten blood samples (3 male and 7 female) taken from subjects who consumed the DI, SM, PFO-SM, and BS-SM drinks for 30 days. Data are expressed as mean ± SD values. Accordingly, ^a^
*p* values of <0.05 were considered significant when compared with the baseline data (D0) of the same group using a parametric paired Student’s *t*-test; ^d^
*p* values of <0.05 were considered significant when comparisons were made between changes in the 4 groups (in brackets) using the non-parametric Kruskal-Wallis test. Abbreviations: BS-SM = black sesame-soybean milk, DI = deionized water, F = female, M = male, PBMC = peripheral blood mononuclear cells, PFO-SM = perilla fruit oil-soybean milk, SD = standard deviation, SM = soybean milk.

**Figure 3 nutrients-14-01721-f003:**
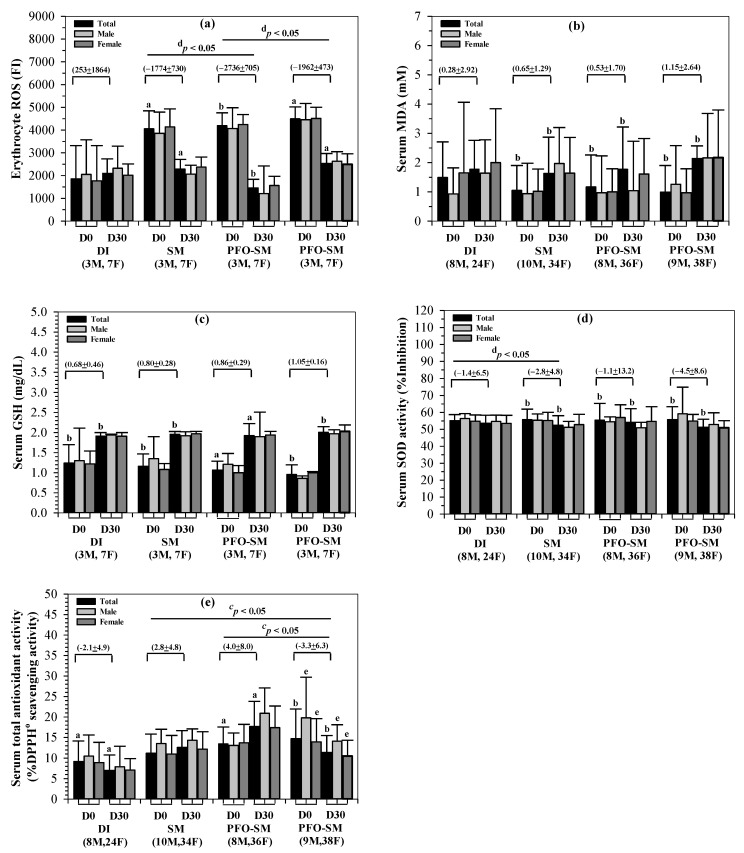
Levels of erythrocyte ROS (**a**), serum MDA (**b**), GSH (**c**), SOD activity (**d**), and TAC (**e**) in subjects who consumed the DI, SM, PFO-SM, and BS-SM drinks for 30 days are shown. Accordingly, ^a^
*p* values of <0.05 were considered significant when compared with the baseline data (D0) of the same group using a parametric paired Student’s *t*-test; ^b^
*p* values of <0.05 were considered significant when compared with baseline data (D0) of the same group using non-parametric Wilcoxon test; ^c^
*p* values of <0.05 were considered significant when comparisons were made between changes of all four groups (in brackets) using parametric one-way ANOVA followed by post hoc Tukey’s HSD or Dunnett T3 tests; and ^d^
*p* values of <0.05 were considered significant when comparisons were made between changes of the four groups (in brackets) using the non-parametric Kruskal-Wallis test; ^e^
*p* values of < 0.05 were considered significant when comparisons were made between male and female subjects in the same group and at the same time using a parametric independent sample of the Student’s *t*-test. Abbreviations: DI = deionized water, DPPH = 2,2-diphenyl-1-picrylhydrazyl, F = female, FI = fluorescence intensity, GSH = reduced glutathione, M = male, MDA = malondialdehyde, PFO-SM = perilla fruit oil-soybean milk, ROS = reactive oxygen species, SD = standard deviation, SM = soybean milk, SOD = superoxide dismutase, TAC = total antioxidant capacity.

**Table 1 nutrients-14-01721-t001:** Nutritional data of SM, 1% PFO-SM, and BS-SM drinks.

Categories	Drink Samples		
	SM	1% PFO-SM	BS-SM
Serving volume (mL)	180	180	180
Total calories (kcal)	90	90	100
Calories from fat (kcal)	15.0	10.0	22.5
Total fat (g)	1.5	1.0	2.5
Saturated fat (g)	0.00	0.50	NA
Trans fat (g)	ND	ND	NA
Cholesterol (mg) ^&^	<1.62	<1.62	NA
Monounsaturated fats (mg)	NA	NA	NA
Polyunsaturated fats (mg)	1.43	1.73	NA
Total carbohydrate (g)	15.0	16.0	14.0
Dietary fiber (g)	0.86	0.18	1.00
Sugars (g)	13.0	12.0	4.0
Protein (g)	5.0	4.0	6.0
Sodium (mg)	20.0	25.0	80.0
Vitamin A (µg)	ND	ND	NA
Vitamin B1 (mg) ^^^	0.060	<0.025	NA
Vitamin B2 (mg) ^^^	<0.025	<0.025	NA
Calcium (mg)	45.81	29.92	NA
Iron (mg)	0.70	0.54	NA

^&^ LOD = 1.62 mg/100 g, ^^^ LOD = 4 μg/100 g. Abbreviations: BS-SM = black sesame-soybean milk, NA = not available, ND = not detected, PFO-SM = perilla fruit oil-soybean milk, SM = soybean milk.

**Table 2 nutrients-14-01721-t002:** Demographic data of healthy volunteers who assessed their level of satisfaction of the 1% PFO-SM drink for 30 d.

Information	1% PFO-SM Product
Gender (*n*)	100 (52F, 48M)
Age range (y)	18–55
Profession (%):	
Student	70
Lecturer	15
Back officer	10
Others	5
Price preference (%):	
15–20 baht	88
21–25 baht	12
Product intension (%):	
Purchased	68
Not purchased	15
Not sure	17
Buying reason:	
Price	65
Taste	24
Brand	3
Health product	8
Other	0

Abbreviations: F = female, M = male, *n* = number, PFO-SM = perilla fruit oil-soybean milk.

**Table 3 nutrients-14-01721-t003:** Sensory acceptance test conducted in the subjects with respect to 0.5% and 1% (*v*/*v*) PFO-SM drinks (*n* = 100 each) using a 9-point hedonic scale. Data are expressed as mean ± SD values.

Characteristics	Sensory Evaluation	
	0.5% PFO-SM	1% PFO-SM
Color	7.28 ± 1.51 ^ns^	7.37 ± 1.51 ^ns^
Flavor:		
PFO odor	5.83 ± 1.63 ^ns^	5.93 ± 1.81 ^ns^
SM odor	6.19 ± 1.67 ^ns^	6.10 ± 1.55 ^ns^
Taste:		
Sweet taste	6.34 ± 1.97 ^ns^	6.34 ± 1.87 ^ns^
SM taste	5.76 ± 1.98 ^ns^	5.97 ± 1.83 ^ns^
PFO taste	6.13 ± 1.77 ^ns^	6.17 ± 1.80 ^ns^
Texture	6.33 ± 1.83 ^ns^	6.17 ± 1.93 ^ns^
Overall acceptance	6.39 ± 1.84 ^ns^	6.29 ± 1.95 ^ns^

Abbreviations: ns = not significant, PFO = perilla fruit oil, PFO-SM = perilla fruit oil-soybean milk, SM = soybean milk.

**Table 4 nutrients-14-01721-t004:** Information and health outcomes of subjects enrolled in this study. Participants consumed DI, SM, PFO-SM, or BS-SM twice daily for 30 days. BW, BMI, SBP, and DBP values were then recorded. Data are expressed as mean ± SD values.

Health Index	DI (*n* = 48)		SM (*n* = 48)		PFO-SM (*n* = 48)		BS-SM (*n* = 48)	
	D0	D30	D0	D30	D0	D30	D0	D30
Gender	14M, 34F	14M, 34F	10M, 38F	10M, 38F	9M, 39F	9M, 39F	18M, 30F	18M, 30F
Age (y)	33.4 ± 12.1	33.4 ± 12.1	33.2 ± 10.1	33.2 ± 10.1	33.6 ± 13.7	33.6 ± 13.7	33.4 ± 12.2	33.4 ± 12.2
BW (kg)	60.6 ± 13.4	60.7 ± 13.3	58.8 ± 12.1	60.2 ± 12.8	58.0 ± 11.4	58.9 ± 11.8	59.3 ± 15.0	59.4 ± 15.2
BMI (kg/m^2^)	20.5 ± 4.2	20.1 ± 4.3	23.3 ± 4.0	23.8 ± 4.3	22.6 ± 3.9	23.0 ± 4.3	23.5 ± 4.7	23.6 ± 4.8
SBP (mm Hg)	124 ± 19	119 ± 15	118 ± 11	115 ± 10	119 ± 15	116 ± 13	122 ± 16	120 ± 17
DBP (mm Hg)	74 ± 10	69 ± 10	71 ± 9.0	68 ± 10	70 ± 10	67 ± 9	71 ± 11	70 ± 11

Abbreviations: BW = body weight, BMI = body mass index, BS-SM = black sesame-soybean milk, DBP = diastolic blood pressure, DI = deionized water, F = female, M = male, PFO-SM = perilla fruit oil-soybean milk, SBP = systolic blood pressure, SD = standard deviation, SM = soybean milk.

**Table 5 nutrients-14-01721-t005:** (**a**) Complete blood count values of subjects who received drinks for 30 days. RBC, WBC, and platelet indices. Data are expressed as mean ± SD values. Accordingly, ^a^ *p* values of <0.05 were considered significant when compared with the baseline (D0) data of the same group using the parametric paired Student’s *t*-test. ^b^ (**b**). Complete blood count values of male and female subjects who received drinks for 30 days. RBC, WBC, and platelet indices. Data are expressed as mean ± SD values of total subjects, as well as for female and male subjects. Accordingly, ^e^
*p* values of <0.05 were considered significant when comparisons were made between male and female subjects in the same group and at the same time using a parametric independent sample of the Student’s *t*-test; ^f^
*p* values of <0.05 were considered significant when comparisons were made between male and female subjects in the same group and at the same time using non-parametric Mann-Whitney U test; ^g^
*p* values of <0.05 were considered significant when comparisons were made between value changes in the same gender following consumption of the drinks using non-parametric Mann-Whitney U test.

(**a**)
**Parameter**	**DI (6M, 34F)**	**SM (10M, 33F)**	**PFO-SM (8M, 37F)**	**BS-SM (9M, 37F)**
**D0**	**D30**	**D0**	**D30**	**D0**	**D30**	**D0**	**D30**
RBC								
(×10^6^/mm^3^)	4.9 ± 0.7	4.7 ± 0.6	4.9 ± 0.6 ^a^	4.8 ± 0.7 ^a^	4.9 ± 0.6	4.9 ± 0.6	4.9 ± 0.5	4.8 ± 0.6
Hb (g/dL)	12.9 ± 2.5	12.9 ± 1.4	13.0 ± 1.5 ^a^	12.8 ± 1.4 ^a^	12.7 ± 1.4	12.7 ± 1.4	13.5 ± 1.5	13.4 ± 1.5
Hct (%)	40.5 ± 3.9	39.6 ± 3.9	40.0 ± 4.3 ^a^	39.3 ± 3.9 ^a^	39.2 ± 3.9	39.2 ± 4.0	41.4 ± 4.7	40.9 ± 4.0
MCV (fL)	83.3 ± 9.8	84.3 ± 10.5	82.8 ± 10.1	82.6 ± 10.1	80.7 ± 9.9	80.7 ± 9.7	83.6 ± 11.3	85.1 ± 1.7
MCH (pg)	27.2 ± 3.7	27.7 ± 3.8	27.1 ± 3.9	26.9 ± 3.8	26.1 ± 3.6	26.8 ± 5.8	27.7 ± 3.0	27.7 ± 3.0
MCHC (g/dL)	32.6 ± 0.9	32.7 ± 0.8	32.6 ± 0.9	32.5 ± 0.8	32.4 ± 0.7	32.3 ± 0.8	32.6 ± 1.0	32.6 ± 1.8
RDW (fL)	15.0 ± 4.0	14.9 ± 3.7	14.4 ± 1.6	14.2 ± 1.7	14.9 ± 2.5	14.8 ± 2.2	13.9 ± 1.2	13.8 ± 1.2
WBC								
(×10^3^/mm^3^)	7.3 ± 1.5	7.1 ± 1.7	7.4 ± 1.3	7.2 ± 1.3	6.8 ± 1.4	6.6 ± 1.7	7.5 ± 2.0	7.2 ± 1.9
Neu (%)	53.8 ± 8.8	51.2 ± 8.9	52.7 ± 9.0	53.4 ± 9.5	54.2 ± 7.1	52.5 ± 8.3	55.0 ± 7.4	53.7 ± 7.5
Lym (%)	35.2 ± 8.4	36.7 ± 7.6	36.7 ± 7.8	36.2 ± 8.8	35.4 ± 6.4	37.0 ± 7.5	34.1 ± 6.1	34.9 ± 6.0
Mon (%)	7.3 ± 2.0	7.0 ± 1.6	6.7 ± 1.3	7.0 ± 1.5	7.3 ± 1.7	7.5 ± 2.0	7.5 ± 1.6	7.6 ± 2.1
Eos (%)	3.1 ± 3.1	4.5 ± 8.7	3.2 ± 3.2	2.8 ± 2.1	2.6 ± 1.8	2.5 ± 1.5	2.9 ± 1.7	3.3 ± 2.5
Bas (%)	0.5 ± 0.2	0.5 ± 0.2	0.5 ± 0.2	0.5 ± 0.2	0.6 ± 0.2	0.5 ± 0.2	0.5 ± 0.2	0.5 ± 0.1
PLT								
(×10^4^/mm^3^)	26.8 ± 5.8	28.0 ± 5.1	25.8 ± 6.2	25.8 ± 5.7	28.4 ± 7.6	28.3 ± 6.9	25.5 ± 5.3	25.2 ± 5.5
MPV (fL)	9.0 ± 0.7	8.8 ± 0.7	8.9 ± 1.0	9.1 ± 1.2	8.9 ± 0.8	8.7 ± 0.8	9.3 ± 1.0	9.1 ± 0.9
PDW	16.4 ± 0.6	18.3 ± 10.8	17.6 ± 7.2	16.7 ± 0.6	16.5 ± 0.6	17.4 ± 6.3	16.4 ± 0.6	16.4 ± 0.5
Pct (%)	0.24 ± 0.05	0.24 ± 0.04	0.23 ± 0.05	0.23 ± 0.05	0.25 ± 0.08	0.25 ± 0.06	0.21 ± 0.01	0.20 ± 0.01
(**b**)
**Parameter**	**DI (6M)**	**DI (34F)**	**SM (10M)**	**SM (33F)**	**PFO-SM (8M)**	**PFO-SM (37F)**	**BS-SM (9M)**	**BS-SM (37F)**
**D0**	**D30**	**D0**	**D30**	**D0**	**D30**	**D0**	**D30**	**D0**	**D30**	**D0**	**D30**	**D0**	**D30**	**D0**	**D30**
RBC																
(×10^6^/mm^3^)	5.1 ± 1.1	4.9 ± 0.5	4.8 ± 0.6	4.6 ± 0.6	5.4 ± 0.6 ^e^	5.2 ± 0.7 ^f^	4.7 ± 0.6 ^e^	4.7 ± 0.6 ^f^	5.3 ± 0.6 ^e^	5.6 ± 0.7 ^f^	4.8 ± 0.5 ^e^	4.8 ± 0.5 ^e^	5.3 ± 0.6 ^e^	5.6 ± 0.7 ^f^	4.8 ± 0.5 ^e^	4.8 ± 0.5 ^f^
Hb (g/dL)	13.3 ± 1.5	13.9 ± 1.6 ^e^	12.5 ± 2.8	12.6 ± 1.3 ^f^	14.6 ± 1.0 ^e^	14.1 ± 0.9 ^f^	12.6 ± 1.4 ^e^	12.4 ± 1.3 ^f^	14.0 ± 1.2 ^e^	14.4 ± 0.9 ^f^	12.4 ± 1.3 ^e^	12.3 ± 1.3 ^f^	15.7 ± 0.7 ^e^	15.7 ± 0.6 ^f^	13.0 ± 1.1 ^e^	12.8 ± 1.0 ^f^
Hct (%)	40.6 ± 4.8	42.5 ± 4.5 ^e^	40.0 ± 3.1	38.6 ± 3.3 ^f^	44.8 ± 2.2 ^e^	43.5 ± 2.2 ^f^	38.5 ± 3.7 ^e^	38.0 ± 3.3 ^f^	43.1 ± 3.3 ^e^	44.7 ± 2.0 ^f^	38.4 ± 3.5 ^e^	38.1 ± 3.3 ^f^	48.2 ± 3.2 ^e^	47.1 ± 1.9 ^f^	39.7 ± 3.2 ^e^	39.4 ± 2.8 ^f^
MCV (fL)	82.0 ± 13.4	87.0 ± 5.3	83.7 ± 8.9	84.1 ± 11.2	83.7 ± 8.6	83.9 ± 8.6	82.5 ± 10.6	82.2 ± 10.6	82.3 ± 11.1	81.2 ± 11.7	80.3 ± 9.8	80.6 ± 9.6	89.1 ± 4.5 ^e^	88.7 ± 4.6	82.2 ± 12.1 ^e^	83.8 ± 8.1
MCH (pg)	26.9 ± 5.1	28.4 ± 2.1	27.3 ± 3.4	27.5 ± 4.2	27.2 ± 3.3	27.2 ± 3.3	27.0 ± 4.1	26.8 ± 4.0	26.7 ± 3.9	26.3 ± 3.9	26.0 ± 3.6	26.9 ± 6.2	29.7 ± 1.7 ^e^	29.5 ± 1.8 ^f^	27.3 ± 3.1 ^e^	27.2 ± 3.1 ^f^
MCHC (g/dL)	32.7 ± 1.0	32.7 ± 0.6	32.6 ± 0.9	32.7 ± 0.9	32.5 ± 0.8	32.4 ± 0.9	32.6 ± 1.0	32.6 ± 0.8	32.3 ± 0.6	32.2 ± 0.7	32.4 ± 0.7	32.3 ± 0.8	32.8 ± 1.6	32.2 ± 0.6 ^f^	32.6 ± 0.8	32.4 ± 0.8 ^f^
RDW (fL)	14.3 ± 1.6	14.5 ± 1.4	15.2 ± 4.4	15.0 ± 1.3	13.9 ± 1.3	13.7 ± 1.3 ^f^	14.5 ± 1.7	14.4 ± 1.8 ^f^	14.0 ± 1.8	14.4 ± 2.2	15.1 ± 2.5	14.9 ± 2.3	13.1 ± 0.8 ^e^	13.1 ± 0.7 ^f^	14.1 ± 1.2 ^e^	14.0 ± 1.2 ^f^
WBC																
(×10^3^/mm^3^)	7.2 ± 0.7	7.1 ± 1.7	7.4 ± 1.7	7.1 ± 1.7	7.4 ± 1.8	7.1 ± 1.3	7.4 ± 1.1	7.3 ± 1.3	8.1 ± 1.5 ^e^	7.6 ± 2.8	6.5 ± 1.2 ^e^	6.4 ± 1.3	8.2 ± 2.2	8.6 ± 2.7	7.3 ± 2.0	6.9 ± 1.6
Neu (%)	55.1 ± 7.7	49.3 ± 9.2	53.4 ± 9.2	51.8 ± 9.0	48.0 ± 6.3	49.2 ± 10.1	54.2 ± 9.3	54.7 ± 9.1	54.6 ± 4.8	53.4 ± 8.6	54.1 ± 7.5	52.3 ± 8.3	52.4 ± 6.4	51.4 ± 11.5	55.6 ± 7.5	54.5 ± 6.3
Lym (%)	32.6 ± 6.7	39.1 ± 4.5	36.0 ± 8.8	36.0 ± 8.2	40.8 ± 5.3	40.3 ± 8.8	35.5 ± 8.0	34.9 ± 8.6	33.4 ± 3.3	34.2 ± 8.2	35.8 ± 6.8	37.6 ± 7.3	34.9 ± 5.6	35.2 ± 7.5	33.9 ± 6.2	34.5 ± 5.7
Mon (%)	8.7 ± 2.1	6.4 ± 1.3 ^g^	6.8 ± 2.0	7.1 ± 1.6 ^g^	6.9 ± 1.1	7.1 ± 1.2	6.6 ± 1.4	7.0 ± 1.6	7.8 ± 1.8	8.4 ± 1.8	7.2 ± 1.7	7.3 ± 2.1	8.4 ± 2.0 ^e^	7.7 ± 2.0	7.2 ± 1.4 ^e^	7.6 ± 2.1
Eos (%)	3.1 ± 2.6	4.1 ± 5.2	3.2 ± 3.2	4.6 ± 9.5	3.7 ± 3.0	2.9 ± 1.9	3.1 ± 3.8	2.8 ± 2.2	3.7 ± 1.3 ^e^	3.6 ± 2.3 ^f^	2.3 ± 1.9 ^e^	2.3 ± 1.2 ^f^	3.6 ± 1.8	5.2 ± 3.9	2.7 ± 1.7	2.8 ± 1.9
Bas (%)	0.5 ± 0.2	0.6 ± 0.2	0.5 ± 0.2	0.5 ± 0.2	0.6 ± 0.2	0.5 ± 0.2	0.5 ± 0.2	0.5 ± 0.2	0.6 ± 0.1	0.5 ± 0.1	0.5 ± 0.2	0.5 ± 0.2	0.5 ± 0.2	0.5 ± 0.2	0.5 ± 0.2	0.5 ± 0.1
PLT																
(×10^4^/mm^3^)	25.6 ± 6.3	30.6 ± 7.6	27.1 ± 5.8	27.3 ± 4.1	25.1 ± 3.3	26.2 ± 3.6	26.0 ± 6.9	25.7 ± 6.2	26.8 ± 6.1	24.0 ± 3.0	28.8 ± 7.9	29.2 ± 7.2	25.2 ± 6.1	26.5 ± 6.8	25.5 ± 5.1	25.0 ± 5.4
MPV (fL)	9.2 ± 1.0	9.1 ± 1.1	9.0 ± 0.6	8.7 ± 0.6	9.0 ± 0.9	9.1 ± 1.2	8.9 ± 1.0	9.1 ± 1.2	8.8 ± 0.6	8.6 ± 0.5	8.9 ± 0.8	8.8 ± 0.9	8.9 ± 1.6	8.7 ± 0.7	9.4 ± 1.1	9.2 ± 0.9
PDW	16.6 ± 0.5	16.6 ± 0.4 ^e^	16.3 ± 0.6	16.2 ± 0.4 ^f^	16.3 ± 0.5	16.7 ± 0.7	16.6 ± 0.6	16.7 ± 0.6	16.1 ± 0.3 ^e^	16.3 ± 0.2	16.6 ± 0.7 ^e^	16.5 ± 0.5	16.5 ± 0.4	16.4 ± 0.4	16.4 ± 0.6	16.4 ± 0.6
Pct (%)	0.23 ± 0.05	0.27 ± 0.05 ^e^	0.24 ± 0.05	0.24 ± 0.03 ^f^	0.22 ± 0.03	0.24 ± 0.05	0.23 ± 0.05	0.23 ± 0.06	0.23 ± 0.05	0.21 ± 0.02 ^f^	0.26 ± 0.08	0.25 ± 0.06 ^f^	0.22 ± 0.05	0.23 ± 0.05	0.24 ± 0.05	0.23 ± 0.05

Abbreviations: In (**a**): Bas = Basophil, BS-SM = black sesamin-fortified soybean milk, DI = deionized water, Eos = Eosinophil, F = female, Hb = hemoglobin, Hct = hematocrit, Lym = Lymphocyte, M = male, MCH = mean corpuscular hemoglobin, MCHC = mean corpuscular hemoglobin concentration, MCV = mean corpuscular volume, Mon = Monocyte, MPV = mean platelet volume, Neu = Neutrophil, Pct = plateletcrit, PDW = platelet distribution width, PLT = platelet, PFO-SM = perilla fruit oil-fortified soybean milk, SM = soybean milk. In (**b**): Bas = Basophil, BS-SM = black sesamin-fortified soybean milk, DI = deionized water, Eos = Eosinophil, F = female, Hb = hemoglobin, Hct = hematocrit, Lym = Lymphocyte, M = male, MCH = mean corpuscular hemoglobin, MCHC = mean corpuscular hemoglobin concentration, MCV = mean corpuscular volume, Mon = Monocyte, MPV = mean platelet volume, Neu = Neutrophil, Pct = plateletcrit, PDW = platelet distribution width, PLT = platelet, PFO-SM = perilla fruit oil-fortified soybean milk, SM = soybean milk.

**Table 6 nutrients-14-01721-t006:** (**a**) Biochemical marker values in the serum of subjects who consumed drinks for 30 days. Data are expressed as values of mean ± SD. Accordingly, ^a^
*p* values of <0.05 were considered significant when comparisons were made with baseline (D0) data of the same group using parametric paired Student’s *t*-test; ^b^
*p* values of <0.05 were considered significant when comparisons were made with baseline (D0) data of the same group using non-parametric Wilcoxon test; ^c^
*p* < 0.05 were considered significant when comparisons were made between changes in the four groups using the parametric one-way ANOVA followed by post hoc Tukey’s HSD or Dunnett T3 tests; ^d^
*p* < 0.05 were considered significant when comparisons were made between changes in the four groups using the non-parametric Kruskal-Wallis test. (**b**) Biochemical marker values in the serum of subjects who consumed drinks for 30 days. Data are expressed as values of mean ± SD. Accordingly, ^e^
*p* values of <0.05 were considered significant when comparisons were made between male and female subjects in the same group and at the same time using a parametric independent sample of the Student’s *t*-test; ^f^
*p* values of <0.05 were considered significant when comparisons were made between male and female subjects in the same group and at the same time using non-parametric Mann-Whitney U test ^g^.

(**a**)
**Parameter**	**DI (6M, 34 F)**	**SM (10M, 33F)**	**PFO-SM (8M, 37F)**	**BS-SM (9M, 37F)**	**DI (6M, 34 F)**	**SM (10M, 33F)**	**PFO-SM (8M, 37F)**	**BS-SM (9M, 37F)**
**D0**	**D30**	**D0**	**D30**	**D0**	**D30**	**D0**	**D30**
BUN (mg/dL)	12.87 ± 3.52	11.75 ± 3.11	13.83 ± 4.55	13.22 ± 4.32	13.09 ± 3.53	12.54 ± 3.57	12.44 ± 4.02	12.36 ± 2.95
CRE (mg/dL)	0.83 ± 0.14	0.88 ± 0.13	0.89 ± 0.19	0.89 ± 0.19 ^d^	0.84 ± 0.11	0.82 ± 0.11 ^d^	0.90 ± 0.16 ^a^	0.83 ± 0.15 ^a,d^
AST (U/L)	11.1 ± 3.5	10.9 ± 3.8	12.1 ± 4.9	11.9 ± 5.1 ^d^	12.3 ± 4.4 ^b^	10.9 ± 4.0 ^b,d^	15.2 ± 8.0 ^b^	12.6 ± 7.1 ^b,d^
ALT (U/L)	5.0 ± 2.0 ^a^	4.1 ± 2.5 ^a^	4.5 ± 2.1	4.6 ± 2.7	4.6 ± 2.3	4.5 ± 3.1 ^d^	5.7 ± 6.5 ^b^	6.2 ± 5.1 ^b,d^
ALP (U/L)	49.9 ± 14.6 ^b^	53.3 ± 12.8 ^b^	50.1 ± 12.9	50.0 ± 13.0 ^d^	51.5 ± 13.1	52.7 ± 13.4 ^d^	54.2 ± 15.7 ^b^	49.2 ± 15.2 ^b,d^
TC (mg/dL)	187 ± 35	197 ± 34 ^d^	183 ± 29 ^b^	213 ± 46 ^b,d^	161 ± 38 ^b^	182 ± 32 ^b,d^	193 ± 34 ^a^	258 ± 33 ^a,d^
TG (mg/dL)	129 ± 56 ^a^	112 ± 54 ^a,c^	99 ± 42 ^b^	113 ± 41 ^b,c^	100 ± 45	89 ± 41 ^c^	94 ± 56 ^b^	153 ± 55 ^b,c^
HDL-C (mg/dL)	68 ± 19	62 ± 32	65 ± 12 ^a^	60 ± 8 ^a^	64 ± 11 ^b^	61 ± 11 ^b^	68 ± 14 ^a^	61 ± 10 ^a^
LDL-C (mg/dL)	98 ± 29	116 ± 58 ^c^	96 ± 26 ^a^	132 ± 41 ^a,c^	77 ± 33 ^a^	105 ± 30 ^a,c^	106 ± 34 ^a^	166 ± 26 ^a,c^
(**b**)
**Parameter**	**DI (8M)**	**DI (32F)**	**SM (9M)**	**SM (35F)**	**PFO-SM (8M)**	**PFO-SM (35F)**	**BS-SM (9M)**	**BS-SM (39F)**
**D0**	**D30**	**D0**	**D30**	**D0**	**D30**	**D0**	**D30**	**D0**	**D30**	**D0**	**D30**	**D0**	**D30**	**D0**	**D30**
BUN (mg/dL)	14.63 ± 4.24	13.16 ± 3.36	12.28 ± 3.13	11.28 ± 2.94	17.97 ± 5.22 ^f^	16.46 ± 4.12 ^f^	12.61 ± 3.57 ^f^	12.27 ± 3.95 ^f^	13.11 ± 3.81	14.59 ± 1.48	12.61 ± 3.57	12.08 ± 3.75	14.50 ± 4.13	14.38 ± 3.27 ^e^	11.95 ± 3.89	11.88 ± 2.70 ^e^
CRE (mg/dL)	0.87 ± 0.21	0.95 ± 0.15	0.82 ± 0.10	0.86 ± 0.12	1.13 ± 0.11 ^e^	1.15 ± 0.11 ^e^	0.85 ± 0.12 ^e^	0.82 ± 0.13 ^e^	0.93 ± 0.07 ^e^	0.94 ± 0.07 ^e^	0.82 ± 0.10 ^e^	0.80 ± 0.10 ^e^	1.10 ± 0.14 ^e^	1.00 ± 0.19 ^e^	0.85 ± 0.13 ^e^	0.80 ± 0.11 ^e^
AST (U/L)	13.4 ± 2.9 ^e^	12.8 ± 4.5	10.3 ± 3.4 ^e^	10.4 ± 3.4	13.9 ± 4.0	15.0 ± 7.6 ^e^	11.6 ± 5.0	11.0 ± 3.8 ^e^	13.4 ± 3.0	12.2 ± 4.1	12.1 ± 4.6	10.5 ± 4.0	27.1 ± 9.9 ^f^	22.5 ± 11.1 ^f^	12.4 ± 4.0 ^f^	10.2 ± 2.7 ^f^
ALT (U/L)	6.3 ± 3.0 ^f^	4.5 ± 3.5	4.4 ± 1.3 ^f^	4.1 ± 2.1	5.5 ± 2.1	5.8 ± 2.3	4.2 ± 2.0	4.2 ± 2.7	5.6 ± 1.8 ^f^	7.8 ± 4.5 ^f^	4.4 ± 2.4 ^f^	3.7 ± 2.1 ^f^	11.8 ± 11.5 ^f^	11.3 ± 8.7 ^f^	4.2 ± 3.5 ^f^	5.0 ± 2.9 ^f^
ALP (U/L)	55.5 ± 13.8	60.3 ± 9.5	47.7 ± 14.5	51.0 ± 12.8	53.4 ± 9.6	53.1 ± 9.5	49.1 ± 13.6	49.1 ± 13.8	61.0 ± 15.8	63.9 ± 14.7 ^e^	49.4 ± 11.7	50.3 ± 11.5 ^e^	63.0 ± 17.6 ^f^	57.8 ± 15.8	52.1 ± 14.7 ^f^	47.2 ± 14.6
TC (mg/dL)	192 ± 36	202 ± 30	185 ± 35	195 ± 35	184 ± 24	200 ± 55	183 ± 31	216 ± 44	149 ± 21	182 ± 26	164 ± 40	181 ± 34	201 ± 27	277 ± 35	191 ± 35	254 ± 32
TG (mg/dL)	137 ± 30	125 ± 41	126 ± 63	107 ± 57	100 ± 32	103 ± 53	98 ± 45	116 ± 39	127 ± 54	133 ± 67 ^f^	94 ± 41	79 ± 27 ^f^	129 ± 74 ^f^	186 ± 85	86 ± 50 ^f^	146 ± 45
HDL-C (mg/dL)	60 ± 57	57 ± 22	70 ± 21	64 ± 35	61 ± 11	53 ± 6 ^e^	66 ± 13	62 ± 7 ^e^	53 ± 4 ^e^	53 ± 3	70 ± 14 ^e^	62 ± 10	58 ± 13	59 ± 12	70 ± 14	62 ± 10
LDL-C (mg/dL)	107 ± 34	129 ± 53	95 ± 27	112 ± 60	95 ± 27	124 ± 49	95 ± 27	134 ± 39	66 ± 19	101 ± 26	80 ± 35	106 ± 31	117 ± 20	177 ± 24	104 ± 35	164 ± 26

Abbreviations: In (**a**): ALT = alanine aminotransferase, AST = aspartate aminotransferase, ALP = alkaline phosphatase, BS-PFO = black sesame-soybean milk, BUN = blood urea nitrogen, CRE = creatinine, DI = deionized water, F = female, HDL-C = high-density lipoprotein-cholesterol, LDL-C = low-density lipoprotein-cholesterol, M = male, PFO-SM = perilla fruit oil-soybean milk, SD = standard deviation, SM = soybean milk, TC = total cholesterol, TG = triglyceride. In (**b**): ALT = alanine aminotransferase, AST = aspartate aminotransferase, ALP = alkaline phosphatase, BS-PFO = black sesame-soybean milk, BUN = blood urea nitrogen, CRE = creatinine, DI = deionized water, F = female, HDL-C = high-density lipoprotein-cholesterol, LDL-C = low-density lipoprotein-cholesterol, M = male, PFO-SM = perilla fruit oil-soybean milk, SD = standard deviation, SM = soybean milk, TC = total cholesterol, TG = triglyceride.

## Data Availability

Not applicable.

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
