# Peer review of "Perilla Fruit Oil-Fortified Soybean Milk Intake Alters Levels of Serum Triglycerides and Antioxidant Status, and Influences Phagocytotic Activity among Healthy Subjects: A Randomized Placebo-Controlled Trial"

_nutrients, 2022, doi:10.3390/nu14091721_

Round 1
Reviewer 1 Report
The present work was designed to develop perilla fruit oil (PFO)-fortified soybean milk (PFO-SM), identify its sensory acceptability, and evaluate its health outcomes. The experiments were carried out properly. However, some questions should be clarified before the paper published.
- The article only mentioned that Perilla fruit oil and soy milk can change the serum triglyceride and other active functions, but did not point out which components have the effect. Is there a synergistic effect between Perilla fruit oil and soy milk on antioxidation and phagocytic activities?
- In the sample treatment part, the soybean was mixed at 60-70℃ and boiled at 80℃ for 15Min. During the boiling process, whether the soybean protein structure changed? Is it inactivated? Do protein components in processed soymilk play a role in antioxidant and phagocytic activity?
3.When choosing to add 0.5% and 1% Perilla fruit oil, should the proportion (concentration) be clarified according to the toxicity test data of the reference? Or is it based on different proportions of taste?
- Why are SM group and BS-SM group not marked with significance in Figure 3?
- Is there any criterion for selecting the male to female ratio in the test population? Why do women outnumber men by four to one?
- The first paragraph of the discussion and the Introduction are very repetitive, so I suggest some modifications.
7.Line523,“A previous study“should be changed to”Some previous studies“.
Reviewer 2 Report
In this manuscript, the authors studied perilla fruit oil (PFO)-fortified soybean milk (PFO-SM) and evaluated its health outcomes. Although the authors run their study on healthy subjects, they recommended the consumption of PFO-SM drinks for the reduction of many chronic diseases. There are two major concerns in this manuscript:
1) No control group (C) without any treatment in the study. The authors studied only three groups: SM, BS-SM, and PFO-SM groups.
2) Volunteers were made up of men and women aged 20-60 years old who were not infected with any serious diseases. The difference in age and sex will affect the different parameters.
The rows in Table 5 did not fit with the parameters. Three separate tables are named Table 5.
Round 2
Reviewer 1 Report
The authors have revised their manuscript accordingly.
Reviewer 2 Report
The authors were successful in responding to comments 1 and 3, but not to comment 2: "we believe that the difference in gender would not have significantly affected the outcomes of either parameter." Please separate male and female parameters in separate tables and provide references to back up your response.
